

# Detection and Explanation of Spatiotemporal Patterns in Late Cenozoic Palaeoclimate Change Relevant to Earth Surface Processes

Sebastian G. Mutz[1], Todd A. Ehlers[1]

[1] Department of Geosciences, University Tübingen, D-72074 Tübingen, Germany

*Correspondence to*: Sebastian G. Mutz (sebastian.mutz@uni-tuebingen.de)

**Abstract**

Detecting and explaining differences between palaeoclimates can provide valuable insight into climatic tipping points and a useful framework of information for Earth scientists investigating processes that are affected by
climate change over geological time. We apply a combination of multivariate cluster- and discriminant analysis techniques to a set of consistently set-up high-resolution palaeoclimate simulations conducted with the ECHAM5 climate model. A pre-industrial (PI) climate simulation serves as the control experiment, which is compared to a suite of simulations of Late Cenozoic climates, namely a Mid-Holocene (MH, ca. 6.5 ka), Last Glacial Maximum (LGM, ca. 21 ka) and Pliocene (PLIO, ca. 3 Ma) climate. For each of the study regions
(Western South America, Europe and Himalaya-Tibet and South Alaska), differences in climate are subjected to geographical clustering to identify dominant modes of climate change and their spatial extent for each time slice comparison (PI-MH, PI-LGM and PI-PLIO). The selection of climate variables for the cluster analysis is made on the basis of their relevance to Earth surface processes and includes 2m air temperature, 2m air temperature amplitude, consecutive freezing days, freeze-thaw days, maximum precipitation, consecutive wet days,
consecutive dry days, zonal wind speed and meridional wind speed. We then apply a two-class multivariate discriminant analysis to simulation pairs PI-MH, PI-LGM and PI-PLIO to evaluate and explain the discriminability between climates within each of the anomaly clusters. Changes in ice cover create the most distinct and stable patterns of climate change, and create the best discriminability between climates in western Patagonia. The distinct nature of European palaeoclimates is mostly explained by changes in 2m air temperature
(MH, LGM, PLIO), consecutive freezing days (LGM) and consecutive wet days (PLIO). These factors typically contribute 30%-50%, 10%-40% and 10%-30% respectively to climate discriminability. Finally, our results identify regions particularly prone to changes in precipitation-induced erosion and temperature-dependent physical weathering.

**Keywords:** Cenozoic climate, climate change, ECHAM5, Last Glacial Maximum, Mid-Holocene, Pliocene, discriminant analysis, discriminability, Earth surface processes, erosion




## 1. Introduction

In the study of Earth surface processes, gaining new quantitative understanding of the atmosphere's interaction with the Earth's surface through erosional processes is limited by the difficulty of establishing reliable palaeoclimatic context for erosion rate histories. Such context is particularly useful when erosion rates are calculated using techniques such as cosmogenic radionuclides and low-temperature thermochronology [e.g. Schaller et al., 2002; Bookhagen et al., 2005; Moon et al., 2011; Insel et al., 2010; Stock et al., 2009], which

integrate over timescales of $10^3$-$10^{6+}$ years. Despite recognition of the influence of climate on tectonic processes and landscape evolution through erosion [e.g. Whipple et al., 1999; Montgomery et al., 2001; Willett et al., 2006; Whipple, 2009; Deal et al., 2018], erosion rates calculated from geo- and thermochronological archives often have to be interpreted under the assumption of modern climate due to insufficient palaeoclimate data [e.g. Starke et al., 2017]. While proxy-based palaeoclimate reconstructions are able to provide part of this

palaeoclimatic context, GCM's (general circulation models) offer a complementary and integrative approach to palaeoclimate reconstructions that yield a more complete picture of palaeoclimate [e.g. Salzmann et al., 2011; Haywood et al., 2013; Jeffrey et al., 2013] and allow better investigation of atmosphere dynamics by conducting sensitivity experiments [e.g. Takahashi and Battisti, 2007]. This study takes advantage of the benefits of GCM's and quantifies differences between simulated palaeoclimates with regard to variables relevant to Earth surface

processes (e.g., rainfall characteristics, temperature derived quantities, wind speed and direction) in order to allow more refined interpretations of potential climatic drivers for changing rates in Earth surface processes.
The Paleoclimate Modelling Intercomparison Project (PMIP) coordinates palaeoclimate modelling efforts [Kageyama et al. 2018] and provides experiment designs for the Mid-Holocene and the Last Interglacial [Otto-Bliesner et al., 2017], the last millennium [Jungclaus et al., 2017], the Last Glacial Maximum [Kageyama et al.,

2017], and the Pliocene Warm Period [Haywood et al., 2016.], which is part of the Pliocene Model Intercomparison Project (PlioMIP). Despite consistent forcings, experiments carried out with different GCM's and model resolutions yield different results due to GCM specific parameterisation. This study is based on a suite of PMIP-style palaeoclimate experiments [Mutz et al., 2018] conducted with the same GCM (ECHAM5) and resolution, which removes the GCM parameterisation related signal in the differences between simulated

palaeoclimates. This experiment framework comprises climate simulations for the pre-industrial (PI, reference year 1850), Mid-Holocene (MH, ~ 6ka), Last Glacial Maximum (LGM, ~ 21ka) and Pliocene (PLIO, ~ 3Ma). For more effective communication of our methods and results, we separate the PI control simulation from MH, LGM and PLIO in discussion by referring only to the latter three as Late Cenozoic climates. These are time periods over which reconstructed erosion rates typically integrate. Understanding how different these

palaeoclimates are from a modern (pre-industrial) climate with regard to variables that potentially affect erosion rates is essential in any comprehensive and merited interpretation of erosion rates, and ultimately allows for better assessment of the influence of climatic and tectonic controls on erosion. Three questions are addressed in this study:

1. What are the spatial patterns of climate change in comparisons of pre-industrial with Late Cenozoic time

periods?

2. How different were Late Cenozoic palaeoclimates to the pre-industrial climate with regard to variables relevant to Earth surface processes?

3. What constitutes these quantified differences between pre-industrial and Late Cenozoic climates?



The geographical focus of this study lies on 4 regions that are frequently investigated with regard to erosion and
uplift histories: South Alaska, Western South America, Himalaya-Tibet and Europe. The first question is
addressed by conducting a cluster analyses on the differences between pre-industrial and Late Cenozoic
palaeoclimates, which subdivides the 4 study regions into geographical clusters governed by a distinct character
of erosion relevant climate change. Whereas Mutz et al. [2018] apply a similar cluster analysis to describe the
modes of climate variability in each palaeoclimate simulation, the results of this study consist of maps showing
the extent of a particular mode of climate change, and thus provides an overview of climate change over time.
The resulting clusters also serve as suitable masks for values used in the discriminant analyses.

Questions 2 and 3 are addressed by conducting discriminant analyses on subdivisions of the 4 study areas,
which are objectively pre-defined by the aforementioned cluster analyses. The results provide a quantitative
assessment and explanation of differences in climate with regard to variables relevant to Earth surface
processes. The third question is answered by conducting a cluster analyses on the differences between pre-
industrial and Late Cenozoic palaeoclimates, which yields the extent of regions governed by a distinct character
of erosion relevant climate change.

The overarching goal is to provide the Earth surface science community with an overview and quantitative
assessment and explanation of how climate changed in the Late Cenozoic with regard to variables relevant to
Earth surface processes.

## 2. Data and Methods

This section describes the data basis, methods and processing steps. In summary, we apply a cluster analysis that
identifies where surface process relevant aspects of climate change are likely, and a discriminant analysis that
quantifies and explains these changes within the regions identified by clustering. This combination of clustering
and discrimination of climate model simulation results yields 5 different sets of results (Fig. 1): Anomaly maps
for a set of climate variables (in the supplemental material to this manuscript), multivariate anomaly cluster
maps and anomaly cluster characterisations (section 2.2), discrimination scores and a measure of relative
contribution by each climate variable to discriminability (section 2.3).


### 2.1 ECHAM5 simulations

General Circulation Models (GCM's) simulate global climate based on our physical understanding of
atmospheric processes and are primarily used to investigate atmospheric dynamics and contemporary climate
change, but have also been applied to improve our understanding of past climates and Earth system dynamics
[e.g. Kutzbach et al., 1993; Ehlers and Poulsen, 2009; Maroon et al., 2015, 2016, Mutz et al., 2016, 2018].
GCM's have become well established tools in geoscience, as is reflected by the work of the Palaeoclimate
Modelling Intercomparison Project (PMIP) [Bracannot et al., 2012], which adds palaeoclimate related
contributions to the Coupled Model Intercomparison Project (CMIP). Palaeoclimate studies address a range of
different time scales including the last millenium [e.g. Jungclaus et al., 2010], orbital [e.g. Gong et al., 2013;
Lohmann et al., 2013; Pfeiffer and Lohmann, 2016; Wei and Lohmann, 2012; Zhang et al., 2014] and tectonic
time scales [e.g. Knorr et al., 2011; Stepanek and Lohmann, 2012]. Comparisons between palaeoclimate
simulations of different time periods are often complicated by use of different models and inconsistencies in
model setup. GCM specific parameterisations and differences in horizontal and vertical resolution introduce





differences between simulations that are not a result of prescribed forcings. In order to avoid these biases, Mutz
et al. [2018] conducted a suite of GCM simulations conducted with the same GCM (ECHAM5) and resolution.
This GCM simulation framework comprises palaeoclimate experiments for pre-industrial times (reference year
1850), the Mid-Holocene (ca. 6.5 ka), the Last Glacial Maximum (ca. 21 ka) and the Pliocene (ca. 3Ma)
climates. The experiments were conducted at a spectral resolution of T159 (ca. 80km x 80km), with 31 vertical
levels and an output frequency of 1 day. The simulations are based on boundary conditions from coupled
AOGCM transient simulations and paleoenvironmental reconstruction initiatives such as PMIPIII [Abe-Ouchi et
al 2015], GLAMAP [Sarnthein et al. 2003], CLIMAP [CLIMAP project members, 1981], PRISM [Haywood et
al., 2010; Sohl et al., 2009; Dowsett et al., 2010] and BIOME6000 [Prentice et al., 2000; Harrison et al., 2001;
Bigelow et al., 2003; Pickett et al., 2004]. For detailed description of model setups, we refer the reader to Mutz
et. al [2018] and references therein. The high resolution and model consistency across all these time slices has
not been achieved previously in the palaeoclimate modelling community. These GCM experiments therefore
represent a unique, state-of-the-art simulation framework suited for investigations of changes in climate
controlled processes across the Late Cenozoic. The GCM (ECHAM5) is a well-established model in the climate
community, and simulated palaeoclimates are in agreement with other modelling and proxy-based
reconstruction efforts. Mutz et al. [2018] use a present-day simulation to establish confidence in the model, and
compare palaeoclimate estimates to compilations of proxy-based reconstructions for MH and LGM precipitation
over South America and Tibet, revealing an overall satisfactory performance of the GCM. The latitudinal
gradients and magnitude of difference in temperature and precipitation are in good agreement with results of
previous palaeoclimate modelling efforts. We refer the reader to Mutz et al. [2018] for a detailed comparison to
other palaeoclimate simulations and proxy-based reconstructions.


### 2.2 Clustering – Multivariate Anomaly Maps

This study's investigation of differences focusses on 4 regions, which (a) are of interest to the Earth surface and
palaeo-altimetry community, and (b) are feasible to work on given the GCM's limitations [cf. Mutz et al., 2018]:
South Alaska (52°-68°N, 125°-165°W), Western South America (5°-56°S, 60°-80°W), Europe (26°-65°N,
22°W-66°E) and Hiamaya-Tibet (0°-60°N, 40°-120°E). Climate change within each of the investigated region is
neither spatially homogenous, nor is it the same in magnitude everywhere [Mutz et al., 2018]. In the example of
PI-LGM climate change in Europe, southern Norway experiences a strong increase in consecutive freezing days
and strong decrease in 2m air temperature, while continental Europe experiences only mild cooling, but strong
increases in intra-monthly 2m air temperature variability. We refer to these combinations of changing climate
attributes as *modes of (climate) change* in this manuscript. Each of the regions is governed by a number of these
distinct modes of climate change. These intra-regional discrepancies merit an informed and objective
subdivision of each region into geographical subdomains governed by one of these modes of change prior to the
investigation of differences in climate through time. Geographical clustering allows such subdivisions on the
basis of similarities in climate change at different locations within each region. It assesses climate change for
each grid box, calculates its similarity to climate change in other grid boxes, and groups them accordingly. More
specifically, it allows the grouping of elements ($i$), in this case climate model surface grid boxes, by the co-
variability of anomalies of selected climatic element attributes. For each region, the contained elements are
subjected to agglomerative hierarchical clustering, followed by k-means clustering corrections [Mutz et al.,



2016; Paeth, 2004] to address the inherent shortcomings of a pure hierarchical approach. The Mahalanobis
distance [e.g. Wilks, 2011] is used as a measure of similarity (of climate change) between clusters in the entire
procedure. Readers are referred to Mutz et al. [2018] for a more detailed description of the procedure and
aforementioned shortcomings.

The clustering is conducted on basis of the same climatic attributes as the discrimination between climates
through time (section 2.3). In total, $M$ (= 9) element attributes, summarised in table 1, are chosen based on their
relevance to Earth surface processes and feasibility of construction from GCM output: near-surface air
temperature (te2m), intra-monthly near-surface air temperature amplitude (t2am), consecutive freezing days
(csfd), freeze-thaw days (fthd), maximum precipitation (pmax), consecutive wet days (cswd), consecutive dry
days (csdd), zonal near-surface wind speeds (u10) and meridional near-surface wind speeds (v10). For
calculation of consecutive freezing days and freeze-thaw days, a surface temperature of 0°C is taken as a
threshold value. The maximum duration of a wet period, i.e. precipitation exceeding 1mm/day [Zolina et al.,
2010; Zhang et al., 2011] constitutes the cswd attribute. Inversely, the maximum duration of a dry period, i.e.
precipitation failing to exceed the 1mm/day threshold [Zin and Jemain, 2010] constitutes the csdd attribute.
These attribute variables are constructed from each palaeoclimate (MH, LGM and PLIO) and reference
simulation (PI) output. The climate attribute anomalies, which serve as a basis for the clustering, are then
calculated for time slice comparisons PI-MH, PI-LGM and PI-PLIO. Clustering requires an a priori decision on
the number of clusters ($k$) or subdivisions per region. The optimal value of $k$ is not known before clustering, but
can be expected to roughly scale with region size. Therefore, the $k$ parameter is varied from 3-5 for South
Alaska and from 5-8 for the larger regions. The cut off point for the parameter is set once the increase in $k$ no
longer results in a cluster with distinct climatic character, but instead results in a weakened or strengthened
character of an already existing cluster. The results consist of optimal geographical subdivisions (climate
clusters $C_1$-$C_k$) with distinct climatic characters, which are described by mean vectors for climate attribute
anomalies. Every cluster characterising vector has a length of $M$. For each of the time slice comparisons and
clusters, the discriminability and relative contribution to it by each of the $M$ attribute variables is quantified in
the procedure described in section 2.3.

### 2.3 Discrimination – Quantifying and Explaining Anomalies

The multivariate linear discriminant analysis (LMD) [e.g. Wilks, 2011] is a statistical tool that allows the
investigation and explanation of differences between two or more groups with regard to multiple attribute
variables. More specifically, it quantifies the discriminability of the groups and the contribution of each of the
attribute variables to this discriminability. The resulting discrimination model can be applied to objectively
categorise an element with unknown group-affiliation. In this study, the time periods (PI, LGM, MH and PLIO)
are used as groups, and the aforementioned climate variables relevant to Earth surface processes are chosen as
group attributes. Since the focus of this study lies on the assessment of the differences between two specific time
periods with regard to multiple climate variables, the problem is treated as a two-group multivariate case.

The centre piece of the analysis lies in finding a discriminant function that best separates the two groups (or
paleo-climate time slices). This is carried out for each comparison, i.e. each pair of paleo-climate time slices.
This discriminant function can be expressed as a linear combination of the climate attribute variables:

$$Y = \upsilon_0 + \upsilon_1 X_1 + \upsilon_2 X_2 + \ldots + \upsilon_m X_m + \ldots + \upsilon_M X_M \tag{1}$$





where Y is the discriminant function, $X_m$ (m=1 … M) are the climate variables used to assess the differences in
paleo-climates, $\upsilon_m$ (m=1 … M) are the discriminant coefficients associated with each variable. $\upsilon_0$ is a constant
(the y-intercept) that is of no relevance to the goodness of separation and will therefore no longer be mentioned.
In this case, M=9 (te2m, t2am, csfd, fthd, pmax, cswd, csdd, u10, v10). Each variable $X_m$ (cf. table 1) contains
elements $x_{mn}$ (n=1 … N), where N is the number of cluster members. Each element (or grid box) is associated
with a discriminant value $y_n$ described by:

$$y_n = \upsilon_1 x_{1n} + \upsilon_2 x_{2n} + \ldots + \upsilon_m x_{mn} + \ldots + \upsilon_M x_{MN} \tag{2}$$

In other words, the elements $y_n$ (n=1 … N) are projected onto the discriminant axis Y. The problem of finding a
discriminant function that best separates the two groups (or paleo-climates) can therefore also be seen as the
process of finding an axis on which the frequency distributions of the projected
elements $y_n$ for the two groups show the smallest overlap. Since overlap is a function of distance between
groups (**D**) as well as the scatter within them (**S**), the difference between these frequency distributions is
described by the distance between the two group centroids (i.e. the group means of the projected elements $y_n$ on
the discriminant axis) and the scatter within the group (i.e. sum of squared deviations from the means), so that

$$\Gamma = \frac{\left(\overline{y_{T1}} - \overline{y_{T2}}\right)^2}{\sum\limits_{j=1}^{n_{T1}} \left(y_{T1j} - \overline{y_{T1}}\right)^2 + \sum\limits_{j=1}^{n_{T2}} \left(y_{T2j} - \overline{y_{T2}}\right)^2} \quad , \tag{3}$$

in a two-group case. $\Gamma$ is the discriminant criterion, T1 and T2 are the two groups (e.g. PI and LGM in the case
of time slice comparison PI-LGM) and $n_{T1}$ and $n_{T2}$ are the number of elements in T1 and T2. In order to find the
best discriminant function Y and corresponding discriminant coefficients $\upsilon_m$ (m=1 … M), $\Gamma$ is maximised, so
that the problem to be solved in the two-group multivariate case of this study can be summarised as

$$\Gamma = \frac{scatter\,between\,groups\,(D)}{scatter\,within\,groups\,(S)} = \frac{\left(\overline{y_{T1}} - \overline{y_{T2}}\right)^2}{\sum\limits_{j=1}^{n_{T1}} \left(y_{T1j} - \overline{y_{T1}}\right)^2 + \sum\limits_{j=1}^{n_{T2}} \left(y_{T2j} - \overline{y_{T2}}\right)^2} \to max\,! \tag{4}$$

In order to solve the problem, we express the discriminant function as a matrix calculation

$$\mathbf{y} = \mathbf{\upsilon_0} + \mathbf{\upsilon X} \tag{5}$$

Where

$$y = \begin{bmatrix} y_1 \\ y_2 \\ \ldots \\ y_N \end{bmatrix} \upsilon_0 = \begin{bmatrix} \upsilon_0 \\ \upsilon_0 \\ \ldots \\ \upsilon_0 \end{bmatrix} \upsilon = \begin{bmatrix} \upsilon_1 \\ \upsilon_2 \\ \ldots \\ \upsilon_M \end{bmatrix} X = \begin{bmatrix} X_{11} & X_{12} & \cdots & X_{1M} \\ X_{21} & X_{22} & \cdots & X_{2M} \\ \vdots & \vdots & \ddots & \vdots \\ X_{N1} & X_{N2} & \cdots & X_{NM} \end{bmatrix} \tag{6}$$

and solve it via partial differentiation. The discriminant coefficients $\mathbf{\upsilon}$ are standardised to yield $\mathbf{\upsilon s}$. The relative
contribution ($\mathbf{\upsilon r}$) of each of the M attribute variables to discriminability is calculated as:



$$\upsilon r_m = \frac{\upsilon s_m}{M} \tag{7}$$

Finally, the skill of the resulting discrimination model is evaluated. For this, the association of each element to groups T1 (PI) or T2 (MH, LGM or PLIO) is forgotten, and the elements are re-categorised according to the critical discriminant values of the models [e.g. Bahrenberg et al., 1992]. The fraction of correct classifications (the **score**) is calculated and used as a measure of "goodness of separation" given the models. The described LMD procedure is applied to each time slice comparison T1-T2 (namely PI-MH, PI-LGM and PI-PLIO) and each of the $k$ climate anomaly clusters ($C_1, C_2, ..., C_k$) in the 4 study regions. Each calculation yields two variables suitable for addressing the problems treated in this study: 1) a measure for **goodness of discriminability (score)** and 2) a measure for the **relative contribution (υr)** of each of the M attribute variables to discriminability. A maximum score of 1 indicates perfect separation of all values, whereas a score of 0 indicates that the discrimination model has no explanatory power at all. Attribute variables associated with a υr value of 1 are solely responsible for the discrimination, whereas those associated with a υr value of 0 contribute nothing to the discriminability between climates.

**2.4 Example Problem**

In summary, the clustering of anomalies (section 2.2) reveals geographical clusters (or subdivisions in each of the study regions), in which similar climate change occurs, and describes the mode of climate change in each of these clusters. The LMD (section 2.3) then quantifies the discriminability of climates in these clusters and explains it with the climatic attribute variables. The set of results for each time slice comparisons and region therefore consists of 4 components: 1) *Anomaly (Cluster) Maps* that show the spatial extent of dominant modes of climate change; 2) *Anomaly Cluster Characterisation* that consists of mean vectors of climate change within each cluster and describe the mode of climate change experienced in the grid boxes assigned to the same cluster; 3) *Discrimination Scores* that describe the goodness of discriminability of climates within each of the anomaly clusters; and 4) *Relative Variable Contribution*, which describes the contribution of each of the climatic attribute variables to the discriminability calculated for each of the anomaly clusters. How these sets of results may be used in answering questions pertaining to climate driven Earth surface processes is demonstrated in the simplified example below.

Erosion rates were calculated, for example by means of cosmogenic nuclides, for a region in a specific area of interest circled on the map (Fig. 2). Although they are taken as modern (time T1) erosion rates, the signal integrates over ten-thousand years and includes erosion rates at time T2. In order to find out if and how significantly erosion rates may actually have been different at time T2, Fig. 2 is consulted. The anomaly cluster map shows the large-scale spatial patterns of changes in climatic variables relevant to Earth surface processes. Each cluster is associated with a specific mode of climate change, and all locations that fall within it experience this type of climate change. The area of interest lies in cluster C1, so all information not related to C1 and the area of interest are shaded pale grey to remove distractions. The purple-green raster plots reveal the type of climate change associated with C1 and thus with circled area of interest: T2 had a little more rainfall, a lot more (consecutive) wet days, higher temperatures, fewer freezing days and fewer (consecutive) dry days. Conditions were warmer and wetter as would be the case for many regions in the Late Pliocene for example. Does the



climate of T2 have a distinct enough signature to tell it apart from T1? The score is reasonably high, which reveals that the climate of T2 does indeed have a distinct signature and consequently, it suggests a good possibility for a different erosional regime or erosive intensity. In order to assess the consequences of the mode of climate change in C1 for erosion, the circles are examined. Those explain which anomaly of which variable is responsible for the distinct signature that has been detected and described above. They indicate that ca. 60% of this "discriminability" can be explained by increases in temperature and ca. 40% by increases in consecutive wet days. Consequently, it is reasonable to assume a priori that erosive processes controlled by these parameters may be affected significantly. The specific sensitivity of conditions and processes at the Earth's surface, e.g. vegetation or critical thresholds in landscape responses, cannot be taken into consideration in these results due to the highly variable nature of it. However, the exact magnitude of those significant (and insignificant) changes are listed in supplemental table T1 to allow the reader to assess the specifics of the impact of climate on processes of the specific area and problem at hand. The above example provides the basic concepts needed to understand the remaining figures in this manuscript.

### 3. Results

This section contains descriptions of results from the cluster- and discriminant analyses carried out for each of the time slice comparisons and study regions. It is divided into sections for the three major study regions: 3.1 Western South America, 3.2 Europe, and 3.3 Himalaya-Tibet. For brevity, the results for South Alaska are included in the supplemental material (S10). For each subsection, regional results for time slice comparisons PI-MH, PI-LGM and PI-PLIO are presented. The figures accompanying each of those consists of four elements (cf. Section 2.4): 1) The first element to the figures are multivariate anomaly maps that were created by clustering the differences between climatic attribute variables in each of these comparisons (cf. Section 2.2) and show the resulting geographical subdivision into clusters governed by a specific mode of climate change. 2) The second element to the figures is (purple and green) raster plots showing the characterisation of those clusters. They describe the mode of change, i.e. the magnitude and direction of changes observed on average in each of the clusters. For visual clarity, the magnitude of change is scaled by the maximum absolute difference in each region and time slice comparison. Numerical values are listed in tabular format in the supplementary material to this manuscript (T1). 3) The third (grey) element consists of the scores for each cluster. These are based on a discriminant analysis carried out for the cluster (cf. Section 2.3) and describe the goodness of discriminability of the palaeoclimates in comparison to the PI control simulation. 4) The fourth and final element to the figures consists of a measure for the relative contribution (υr) of each of the 9 climate variables (table 1) to the overall discriminability between two time slices in each geographical cluster. This element is displayed as a layer of circles on top of element 2 and reveals how much an anomaly of a specific variable (represented by a specific shade of purple or green) contributes to the discriminability between PI and a palaeoclimate in each of the clusters. Larger circle diameters correspond to greater contributions.

For brevity, results for these time slice comparisons PI-MH, PI-LGM and PI-PLIO are simply referred to as results for MH, LGM and PLIO respectively. The notation MH-$C_i$, LGM-$C_i$ and PLIO-$C_i$ is used to refer to results for the $i^{th}$ geographical subdivision (or cluster) in the respective set of results. Description of changes in climate are implicitly discussed in the context of the control simulation and are therefore descriptions of deviations from the PI climate. Discussions of multivariate anomaly maps and cluster characterisation (elements



1 and 2) focus mostly on the stable and persistent patterns seen in the results, i.e. geographical clusters that are least sensitive to changes in *k* and keep their distinct character. Discussions of scores and relative contribution to discriminability ($\upsilon r$) focus primarily on clusters with good discriminability. Additional information on climate change on a sub-cluster scale are given in the form of single-variable anomaly maps in the supplementary material to this manuscript (S1-S9).

### 3.1 Western South America

*Large scale patterns and modes of climate change*

In western South America, the dominant modes of change for the MH are captured in clusters MH-C$_3$, MH-C$_4$, MH-C$_5$ and MH-C$_6$ (Fig. 3 a,d). MH-C$_3$ covers much of central Patagonia and is characterised by decreases in consecutive freeze-thaw days and increases in consecutive freezing days and consecutive wet days. MH-C$_4$ is the mode of change observed in most of Argentina, central and southern Andes. It consists of relatively small increases in consecutive wet days and freeze-thaw days. MH-C$_5$ covers most of the tropics in the region and is characterised by decreases in consecutive wet days and maximum precipitation, and relatively small increases in 2m air temperature and consecutive dry days. Relatively large increases in 2m air temperature and consecutive dry days, and decreases in maximum precipitation and consecutive wet days constitute the mode of change for MH-C$_6$, which extends over low-altitude subtropics in the region.

LGM-C$_1$ – LGM-C$_3$ share a number of characteristics. These modes of changes include decreases and 2m air temperature and increases in 2m air temperature amplitude and consecutive freezing days (Fig. 3 e). Furthermore, the region occupied by LGM-C$_1$ – LGM-C$_3$ is covered by ice in the LGM. Differences between these modes of changes include a large increase in maximum precipitation in LGM-C$_1$ and decreases in consecutive wet days in LGM-C$_2$. LGM-C$_4$, covering much of the subtropics in the region, is characterised by relatively little change in all of the investigated variables. LGM-C$_5$ covers much of eastern Argentina and experiences increases in 2m air temperature amplitude. LGM-C$_6$ covers much of the tropics of the region and is characterised primarily by decreases in maximum precipitation and consecutive wet days.

The dominant modes of change in the PLIO are described by PLIO-C$_1$, PLIO-C$_2$, PLIO-C$_3$ and PLIO-C$_5$ (Fig. 3 f). Covering much of eastern Argentina and parts of the central Andes, PLIO-C$_1$ is characterised primarily by relatively large decreases in consecutive dry days and increases in maximum precipitation and consecutive wet days. The grid boxes assigned to PLIO-C$_2$, extending over most of the low-altitude tropics and subtropics and the Atacama desert, experience very little change on average. For grid boxes assigned to PLIO-C$_3$, a decrease in maximum precipitation and consecutive wet days, and increase in consecutive dry days and meridional wind speeds can be observed. PLIO-C$_5$ extends over much of the central and southern Andes and is characterised by decreases in freeze-thaw days and increases in 2m air temperature and meridional wind speeds. While PLIO-C$_6$ experiences some of the largest changes, it only covers the Northern and Southern Patagonian Ice Fields and coincides with the reduction of the ice cover in the PLIO simulation.

*Discriminability*

The discrimination scores (Fig. 3 d,e,f) are highest for the LGM and lowest for the MH, and changes in temperature, consecutive freezing days, maximum precipitation and consecutive dry days are factors that



explain much of the discriminability overall. LGM-$C_1$ – LGM-$C_3$ s have the highest scores. In all 3 clusters,
decreases in 2m air temperature are one of the primary contributors to the discriminability between LGM and PI
climate. It explains 40%-50% of the discriminability in LGM-$C_1$ and 30%-40% in LGM-$C_2$ and LGM-$C_3$. With
20%-30% explained discriminability in LGM-$C_1$ and 30%-40% explained discriminability in LGM-$C_3$, increases
in consecutive freezing days are a second important factor for discriminability between the climates in western
Patagonia. Discrimination with PLIO and PI simulations yields the highest scores for PLIO-$C_6$, which covers the
Patagonian Ice Fields, and PLIO-$C_5$. Increases in 2m air temperatures and decreases in consecutive wet days and
consecutive dry days explain the discriminability in PLIO-$C_6$ in equal parts (20%-30%). An increase in
temperatures and relatively small decrease in consecutive freezing days explain 20%-30% and 40%-50% of the
discriminability in PLIO-$C_5$ respectively.

**3.2 Europe**

*Large scale patterns and modes of climate change*
MH-$C_1$ covers Spain and the region east of the Caspian sea (Fig. 4 a) and is associated with the least change in
climate attribute variables. MH-$C_2$, covering areas in western Europe, western Russia and the Mediterranean
coasts (Fig. 4 a), experiences an increase in maximum precipitation (Fig. 4 d). Ukraine, Poland, much of the
Baltic Sea coast and southern Scandinavia are assigned to MH-$C_3$ and experience decreases in freeze-thaw days.
MH-$C_4$ and MH-$C_5$ primarily cover northern Africa and are characterised by increases in 2m air temperature and
maximum precipitation.
LGM-$C_1$-LGM-$C_4$ (Fig. 4 b) are all partially characterised by a decrease in 2m air temperature, freeze-thaw days
and increases in consecutive freezing days (Fig. 4 e). It should be noted that grid boxes assigned to these clusters
are covered by the Scandinavian Ice Sheet in the LGM simulation. LGM-$C_5$ extends over much of the
Mediterranean region, Spain and European North Russia and characterised by increases in 2m air temperature
amplitude, consecutive dry days and relatively small increases in freeze-thaw days and decreases in maximum
precipitation and consecutive wet days. Most of central Europe, western Asia and North Africa is assigned to
cluster LGM-$C_6$ and experiences the least change.
The dominant modes of change in the PLIO are captured in PLIO-$C_3$-PLIO-$C_6$ (Fig. 4 c). PLIO-$C_3$ is a mode of
change mostly seen in parts of North Africa and characterised by increases in meridional and zonal wind speeds
(Fig. 4 f). In the coastal regions north of it, very little change is seen in the PLIO (PLIO-$C_4$). PLIO-$C_5$ covers
much of central Europe and experiences decreases in freeze-thaw days and 2m air temperature amplitude, and
relatively small increases in 2m air temperature. European Russia and parts of Scandinavia are assigned to
PLIO-$C_6$ and experience increases in freeze-thaw days and 2m air temperature, and decreases in consecutive dry
days and 2m air temperature amplitude. PLIO-C7 is mostly distributed along parts of the Mediterranean, Black
Sea and Caspian Sea coasts and characterised by increases in consecutive dry days and 2m air temperature, and
a decrease in freeze-thaw days. PLIO-C8 covers southeastern Norway and the Alps and is characterised by
decreases in consecutive freezing days and 2m air temperature amplitude, and increases in 2m air temperature
and freeze-thaw days.

*Discriminability*





In all time slice comparisons, changes in 2m air temperature explains most of the discriminability in many of the geographical clusters (Fig. 4 d,e,f). Changes in consecutive freezing days and consecutive wet days are also major contributors to discriminability in the LGM and PLIO respectively. $LGM-C_1$, $LGM-C_2$, $LGM-C_5$ and $LGM-C_6$ are associated with the highest scores for the LGM. 20%-24% of the discriminability in the clusters can be explained by decreases in temperature, and a similar amount can be explained by increases in consecutive freezing days. Although all PLIO scores are high, the PLIO cluster in central Europe ($PLIO-C_5$) is associated with the highest value. The discriminability in the cluster can be explained by increases in 2m air temperature (30%-40%), increases in consecutive wet days (20%-30%), decreases in consecutive dry days (10-20%) and decreases in temperature amplitude (10-20%). Discriminability in the high-altitude cluster ($PLIO-C_8$) can be explained by increases in consecutive dry days (10-20%) and decreases in consecutive wet days (20-30%), maximum precipitation (20-30%) and temperature amplitude (10-20%).

### 3.3 Himalaya-Tibet

*Large scale patterns and modes of climate change*

The stable patterns for the MH results include $PLIO-C_1$ covering the region south of the Himalayan orogen, $MH-C_2$ covering central India and Southeast Asia, $MH-C_4$ in the region around the Caspian sea, and $MH-C_5$ north of the Caspian and Aral Sea (Fig. 5 a). $MH-C_1$ is characterised by increases in consecutive wet days and maximum precipitation and decreases in consecutive dry days and 2m air temperature (Fig. 5 d). $MH-C_2$ mostly experiences changes in meridional and zonal wind speeds. $MH-C_4$ is characterised by increases in 2m air temperature amplitude and consecutive freezing days and decreases in freeze-thaw days. $MH-C_5$ grid boxes are associated with relatively large increases in freeze-thaw days and smaller increases in consecutive freezing days and 2m air temperature amplitude.

$LGM-C_1$ mostly covers the northernmost parts of the region and the Himalayan orogen (Fig. 5 b). The changes associated with it are decreases in 2m air temperature and increases in 2m air temperature amplitude and consecutive dry days (Fig. 5 e). The modes of change described by $LGM-C_2$ and $LGM-C_3$ govern large parts of the region, including the Arabian Peninsula, Iran, central Asia, the Tibetan Plateau and Tarim Basin, Mongolia and parts of China. These regions experience relatively small decreases in temperature. Central India and eastern Southeast Asia ($LGM-C_4$) are associated with decreases in consecutive wet days, maximum precipitation and zonal wind speeds, and increases in consecutive dry days. Parts of Kazakhstan, southern Russia, China, southeast Asia and northern India are assigned to $LGM-C_5$, which is characterised by increases in consecutive dry days.

The region covered by $PLIO-C_1$ includes northern India along the Himalayan orogen (Fig. 5 c) and experiences decreases in consecutive dry days and increases in consecutive wet days and maximum precipitation (Fig. 5 f). $PLIO-C_2$ covers most of the study region and is associated with relatively little change in all climatic attributes except meridional wind speeds. Central Asia is mostly assigned to $PLIO-C_3$ and experiences an increase in 2m air temperature and decrease in freeze-thaw days. The north of the study region ($PLIO-C_4$) is characterised by decreases in 2m air temperature amplitude and consecutive freezing days and increases in 2m air temperature and freeze-thaw days. Finally, $PLIO-C_5$ mostly covers the high altitude locations of the Himalaya-Tibet region that are close to steep topographic gradients, incl. The Himalayan orogen. This cluster is associated with





decreases in wind speeds, consecutive freezing days, consecutive wet days and 2m air temperature and increases
in consecutive dry days and 2m air temperature.

*Discriminability*

The significance of the climate attributes in explaining the discriminability the Himalaya-Tibet clusters (Fig. 5
d,e,f) is more variable than in Europe. While changes in 2m air temperature are important in most of the MH
and LGM results, there is no clear dominant factor for PLIO clusters. In the LGM, the discriminability in the
high-altitude/high latitude cluster (LGM-$C_1$) is mostly explained by decreases in 2m air temperature (30%-
40%), mild increases in consecutive freezing days (20-30%) and mild decreases in consecutive wet days (10%-
20%). LGM-$C_4$ has the second highest discrimination score, and the discrimination in this cluster is explained
by decreases in consecutive dry days (10%-20%) and increases in consecutive wet days (10%-20%), maximum
precipitation (30%-40%) and 2m air temperature (20%-30%). For PLIO, the type of climate change governing
the largest cluster (PLIO-$C_2$) causes discriminability that is primarily explained by mild decreases in consecutive
freezing days (20%-30%) and mild increases in consecutive wet days (20%-30%) and 2m air temperature
amplitude (10%-20%). Discriminability in the cluster occupying the region south of the Himalayan orogen
(PLIO-$C_1$) is explained by decreases in consecutive dry days (10%-20%) and 2m air temperature (10%-20%),
and increases in consecutive wet days (10%-20%), maximum precipitation (30%-40%) and consecutive freezing
days (10%-20%). Cluster PLIO-$C_5$ is associated with a discriminability best explained by increases in
consecutive dry days (30%-40%) and decreases in maximum precipitation (10%-20%) and consecutive freezing
days (20%-30%).

**4. Discussion**

This section describes method-related features and problems, and highlights commonly occurring patterns of
change, provides possible explanations for those, and discusses these changes in context of erosional processes.

*The role of large scale features*

For many of the LGM and some of the PLIO results, changes in 2m air temperature and/or consecutive freezing
days significantly contribute to the discriminability in clusters covering mid-latitudes. LGM-$C_1$ – LGM-$C_3$ in
South America, LGM-$C_1$ and PLIO-$C_4$ in the Himalaya-Tibet region are examples of this. Many of these high-
latitude clusters are also characterised by large changes in 2m air temperature and 2m air temperature amplitude
in the LGM and PLIO results. The preferential cooling in higher latitudes during the LGM and enhanced
meridional temperature gradient [e.g. Otto-Bliesner et al., 2006; Bracannot et al., 2007; Mutz et al., 2018] can be
expected to result in more pronounced seasonality and thus higher variation in near surface temperature
amplitude. Inversely, the accentuated warming in higher latitude during the (Late) Pliocene [e.g. Salzman et al.,
2011; Ballantyne et al., 2010; Mutz et al., 2018] would result in the opposite. These previously studied large
scale features explain much of the characterisation of high-latitude clusters and the significant contribution of
changes in temperature-related variables to regional discriminability. Associated changes in temperature
variables can have decisive impacts physical weathering due to changes in glacial and periglacial processes (see
below), and biotic weathering by influencing vegetation cover.



*The role of glaciers and periglacial processes*

Changes in temperature in high altitude regions can impact physical weathering through glacial erosion [e.g. Egholm et al., 2009, Herman et al. 2013] and periglacial processes [e.g. Andersen et al., 2015, Marshall et al., 2015]. In southernmost South America, northern Europe and South Alaska (cf. supplemental information), the high discriminability and modes of change on the multivariate anomaly maps for the LGM are primarily controlled by the glaciers covering most of the region. Furthermore, many modes of change in the study regions

involve consecutive freezing days and freeze-thaw days. Changes from ice-free to ice-covered conditions, such as in LGM-$C_1$-LGM-$C_3$ in South America and LGM-$C_1$-LGM-$C_4$ in Europe, are associated with increases in consecutive freezing days and decreases in freeze-thaw days. The opposite is the case for some modes of changes in the PLIO. For example, PLIO-$C_6$ in South America is associated with changes from ice-covered to ice-free conditions, as well as with an increase in consecutive freezing days. It may therefore shift from glacier

to frost-cracking dominated erosional processes. These modes of change in the PLIO mark possible transition from glacier governed processes to periglacial processes and thus increased frost-cracking as the Earth's surface spends more time in the frost-cracking window [e.g. Matsuoka, 2001; Schaller et al., 2002; Andersen et al., 2015, Marshall et al., 2015], whereas many LGM modes of change suggest the opposite. Finally, glacial pre-conditioning of a landscape can modulate the effect of precipitation on landsliding [Moon et al. 2011].


*The role of precipitation characteristics*

Areas that have been covered by glaciers during the LGM and experienced a post-LGM increase in maximum precipitation or consecutive wet days may be particularly prone to precipitation-triggered landslides. This is the case, for example, in the regions covered by LGM-$C_2$ and LGM-$C_3$ clusters in Patagonia and LGM-$C1_1$ – LGM-

$C_3$ in Europe. More generally, changes in storminess affect erosion through river incision and sediment transport [e.g. Whipple and Tucker, 1999; Hobley et al., 2010]. Maximum precipitation and consecutive wet days are measures of storm intensity and duration respectively, which are primary controls for runoff and relevant for erosion. In most cases, such as the results for the Himalaya-Tibet region in the LGM and PLIO, the co-variability of consecutive wet days, maximium precipitation and consecutive dry days is intuitive: changes in

consecutive wet days and maximum precipitation coincide with changes in consecutive dry days in the opposite direction. Even though palaeovegetation is considered in the set-up of the GCM simulations [Mutz et al., 2018], the modulating effect of vegetation on the impact of changes in the precipitation attribute variables on erosion [e.g. Gyssels et al., 2005] cannot be taken into account here, and thus the reader is advised to do so in their evaluation of the effect of these changes on Earth surface processes. In absence of significant vegetation

changes, areas such as Bhutan, Nepal, Bangladesh and parts of Northern India (MH-$C_1$ and PLIO-$C_1$), which experiences strong increases in consecutive wet days and maximum precipitation in the MH and PLIO, are likely to have experienced an increase in such precipitation-induced erosion at these times.

*The role of winds*

Changes in wind speed components affect aeolian erosion, transport, and deposition, as well as mean raindrop trajectories, which should also be taken into consideration in the assessment of local precipitation-induced erosion [de Lima et al, 1992]. The results of this study reveal that changes in near surface meridional and zonal wind speeds contribute little to the discriminability between climates even in regions that experience wind



direction changes due to a different ice cover in Europe [e.g. Siegert and Dowdeswell, 2004], which are reproduced well by the model. Wind speeds only show a significant contribution to discriminability (20%-30%) in the subtropical latitudes of South America due to slower meridional winds in the LGM in the region. The distinctiveness in the character of atmospheric dust transport during the LGM [e.g. Andersen et al., 1998] and thus aeolian erosion may be attributed more to system response and changes in vegetation, which cannot be taken into account in this study, than to a distinctiveness of LGM wind speeds.

*Comments on methodical implications*

PLIO and LGM clusters are more stable than MH clusters on the multivariate anomaly maps. This stability can be attributed to the relatively large magnitude of climate change in PLIO and LGM time slices. Lower variance of MH anomalies make element attribution to anomaly clusters in the MH more sensitive to randomisation and re-categorisation procedures (section 2.2). Consequently, the nature of MH patterns can be seen as the result of climate change of lower magnitude and less distinctiveness. The most important limitation is the poor representation of precipitation amount in areas of high topography and rainfall [Meehl et al., 2007 and comparisons with ERA-interim and station-based observations not presented here]. However, the threshold for what constitutes a "wet day" or "dry day" is relatively low [Zolina et al., 2010; Zhang et al., 2011; Zin and Jemain, 2010], so that the typical overestimation of total precipitation amount by ECHAM5 in such regions has little or no effect on the attribute variables *consecutive wet days* and *consecutive dry days*, particularly when analysed in comparison to another simulation at a different time (as was done here) which helps reduce any systematic model bias towards high precipitation rates. The overall performance of the palaeoclimate simulations is decent, as comparison with proxy-based reconstructions showed [Li et al. 2017; Mutz et al. 2018]. Erosion relevant processes that take place on high spatial or temporal scales, such as intra-storm variations and rainfall characteristics [e.g. Ran et al., 2012], cannot be quantified in this study due to limited model resolution, output frequency and accuracy of such estimates on that scale. The consideration of non-climatic factors, such as local topography, slope and vegetation, is beyond the scope of this study, and the reader is advised to take these into consideration in their assessment of the effect of documented climate change on Earth surface processes.

**5. Conclusions**

In this study, we quantified the differences between pre-industrial and Late Cenozoic palaeoclimates with regard to variables relevant to Earth surface processes, explained these quantified differences and identified dominant patterns and modes of palaeoclimate change. The key findings of this study are:

- LGM and PLIO climate change is more distinct and more easily quantified than climate change of the MH. This is reflected in the stability of geographical regions (clusters) showing the extent of regions governed by distinct modes of climate change.

- Changes in ice cover result in very distinct signatures of climate change. This is reflected by 1) the creation of clusters geographically associated with ice cover changes, 2) the persistence of these



clusters when $k$ is varied in the procedure, and 3) ice cover changes in South America leading to the best discriminability overall.

- In Europe, changes in 2m air temperature explain most of the discriminability between pre-industrial and all three palaeoclimates (MH, LGM and PLIO). Changes in consecutive freezing days and consecutive wet days are also significant contributors to climate discriminability in LGM and PLIO results respectively. Consequently, these factors lend the Late Cenozoic palaeoclimates their unique signature and should be central in assessments of changes in Earth surface processes.

- Increases in freeze-thaw days and temperature often coincide with decreases in consecutive freezing days, and vice versa. Regions governed by these modes of changes, such as western Patagonia during the LGM, are prone to changes in erosional process domain from peri-glacial to glacial or vice versa.

- Increases in consecutive wet days and maximum precipitation often coincide with decreases and consecutive dry days. Regions governed by these modes of change, such as locations south of the Himalayan orogen in the PLIO, can be expected to be particularly prone to changes in erosion induced by precipitation and storm characteristics.

**Data availability:** All data files used in the presentation of results, as well as the model simulation results this study is based on [Mutz et al., 2018], are freely available through the data download section of our work group wiki (esdynamics.geo.uni-tuebingen.de/wiki/) as set of data products tailored to the needs of different geoscientific communities.

**Competing interests:** The authors declare that they have no conflict of interest.

**Acknowledgements:** European Research Council (ERC) Consolidator Grant number 615703 provided support for this study. Additional support is acknowledged from the German science foundation (DFG) project number 365266215.

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



**Figures and Tables**

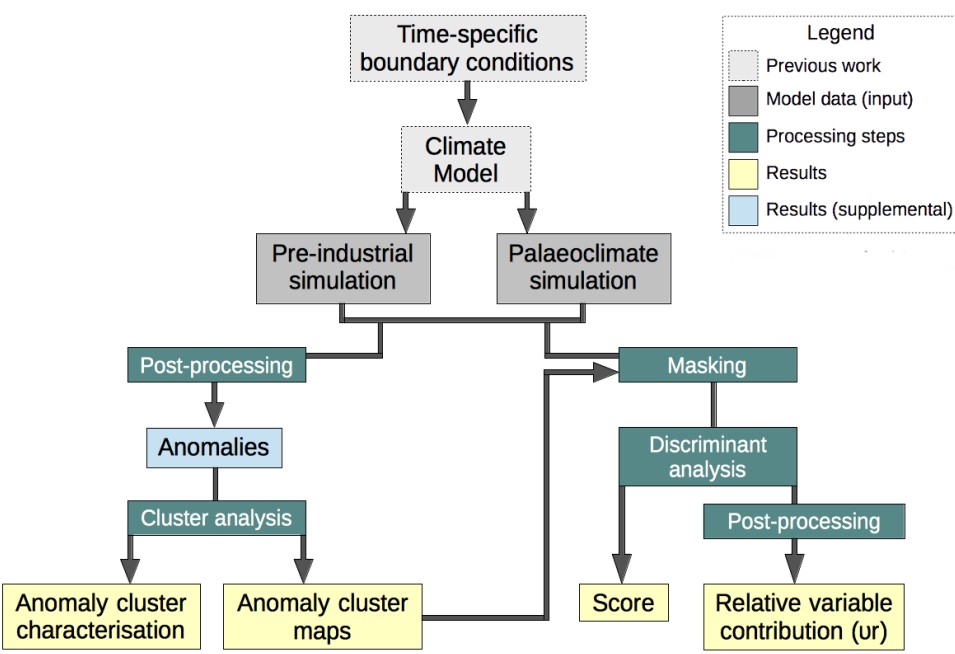

Fig. 1 Anomalies are created from pre-industrial (PI) and Late Cenozoic palaeoclimates (MH, LGM, PLIO). These are subjected to geographical clustering, which results in the identification of distinct modes of climate change (anomaly cluster characterisation) and maps showing the spatial extent of regions governed by these modes (anomaly cluster maps). These are used as geographical masks for palaeoclimate simulations. For each of these anomaly clusters, a discriminant analysis is conducted to quantify the discriminability in each cluster (score) and the relative contribution of each climatic variable to this discriminability.












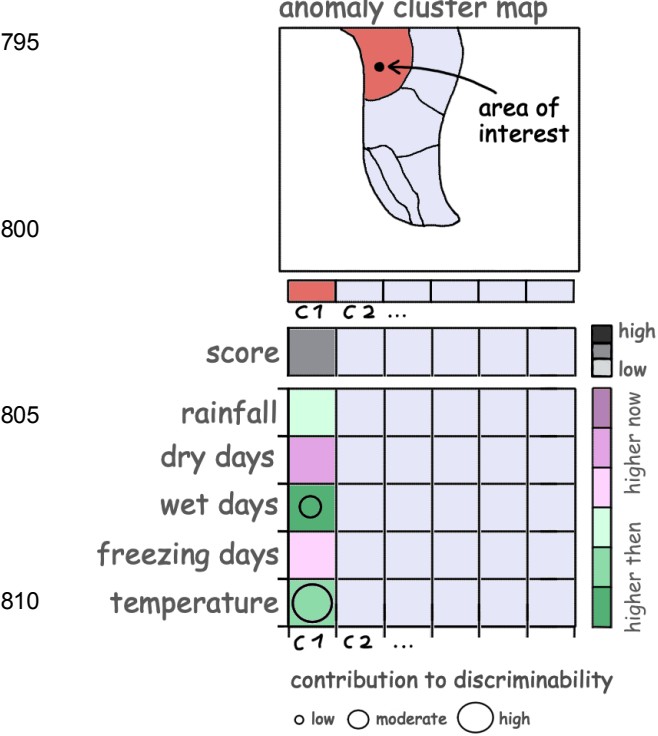

Fig. 2 Example problem: Investigating how climatic boundary conditions for erosion processes have changed in a specific region of interest involves consultation of part of the conceptualised results figure. The location of the region of interest geographically coincides with the region assigned to cluster C1. Consequently, only the results

related to C1 are consulted and the rest is greyed out. The clusters were calculated from the differences in the values (delta values) of geomorphic variables between two different times (T1 and T2) in the late Cenozoic, and thus represent a specific mode of change. The mode of change associated with the cluster of interest (C1) is revealed by the purple-green column. The relative contributions of the delta values to the overall discriminability between T1 and T2 in cluster C1, indicated by the score, is revealed by the diameters of the

circle superimposed onto the delta values in the purple-green column.



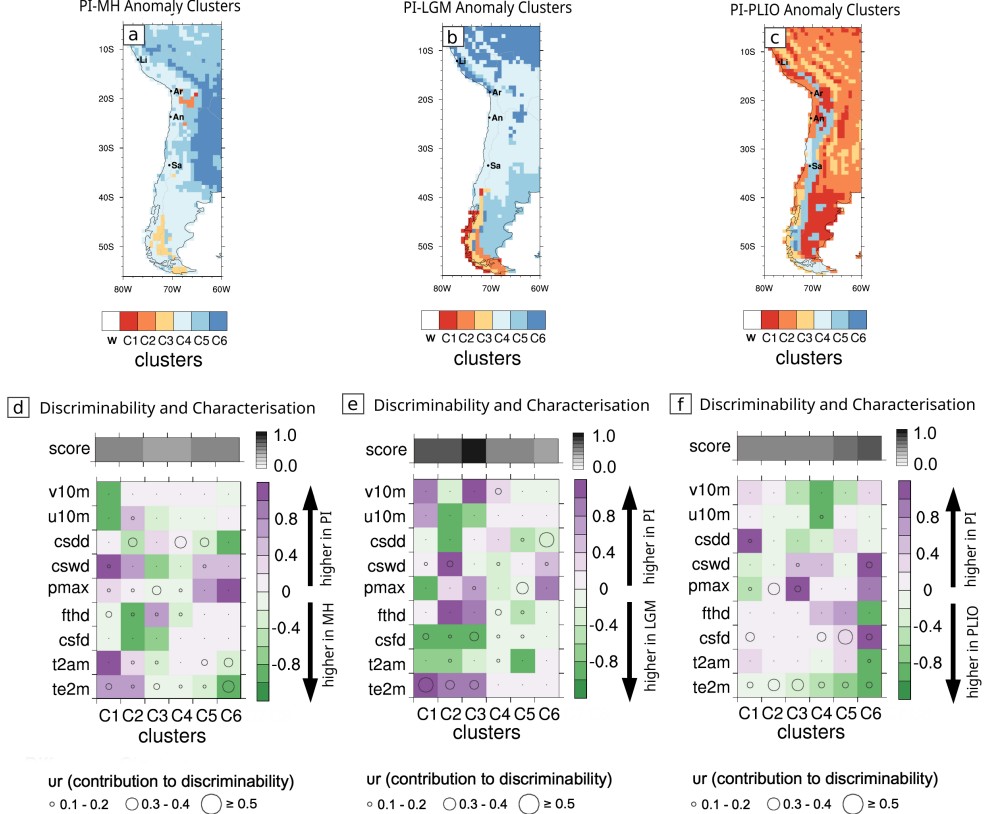

Fig. 3 The multivariate anomaly maps for time slice comparisons PI-MH(a), PI-LGM(b) and PI-PLIO(c) show the geographical coverage of clusters $C_1$-$C_i$ in Western South America, which describe the spatial extend of regions characterised by similar modes of change. The corresponding modes of change (d,e and f) for each cluster are expressed as relative changes in each of the 9 investigated variables (table 1): 2m air temperature (te2m), 2m air temperature amplitude (t2am), consecutive freezing days (csfd), freeze-thaw days (fthd), maximum precipitation (pmax), consecutive wet days (cswd), consecutive dry days (csdd), zonal near surface wind speeds (u10) and meridional near surface wind speeds (v10). The score (d,e and f) expresses the goodness of discriminability between the palaeoclimate pairs PI-MH(d), PI-LGM(e) and PI-PLIO(f) in each of the anomaly clusters. The size of the circles corresponds to the relative contribution of each of the 9 climatic attribute variables to the measured discriminability in each anomaly cluster for all three time slice comparisons.





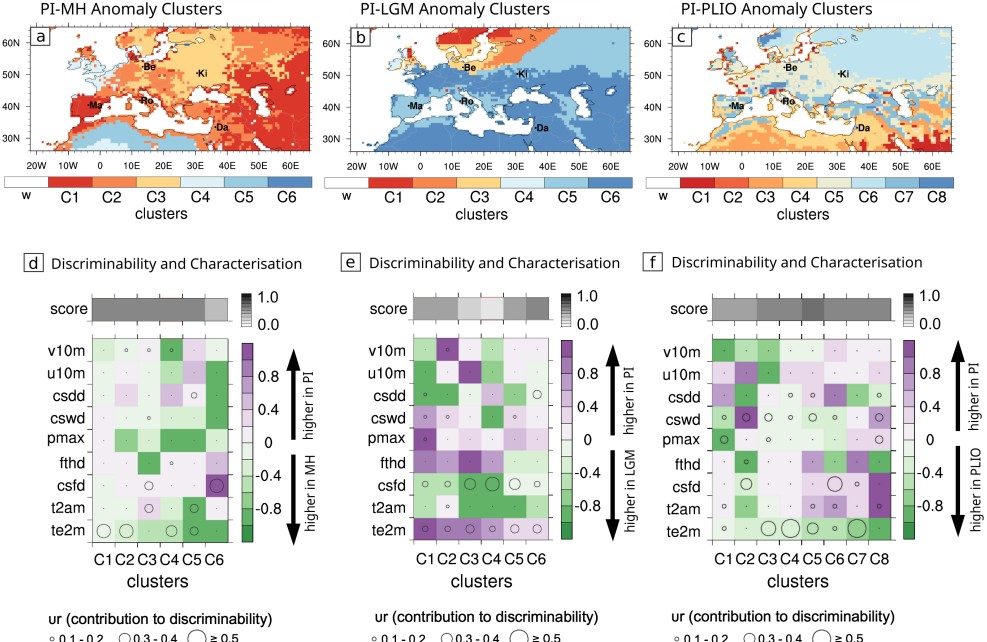

Fig. 4 The multivariate anomaly maps for time slice comparisons PI-MH(a), PI-LGM(b) and PI-PLIO(c) show the geographical coverage of clusters $C_1$-$C_i$ in Europe, which describe the spatial extent of regions characterised by similar modes of change. The corresponding modes of change (d,e and f) for each cluster are expressed as relative changes in each of the 9 investigated variables (table 1): 2m air temperature (te2m), 2m air temperature

amplitude (t2am), consecutive freezing days (csfd), freeze-thaw days (fthd), maximum precipitation (pmax), consecutive wet days (cswd), consecutive dry days (csdd), zonal near surface wind speeds (u10) and meridional near surface wind speeds (v10). The score (d,e and f) expresses the goodness of discriminability between the palaeoclimate pairs PI-MH(d), PI-LGM(e) and PI-PLIO(f) in each of the anomaly clusters. The size of the circles corresponds to the relative contribution of each of the 9 climatic attribute variables to the measured

discriminability in each anomaly cluster for all three time slice comparisons.



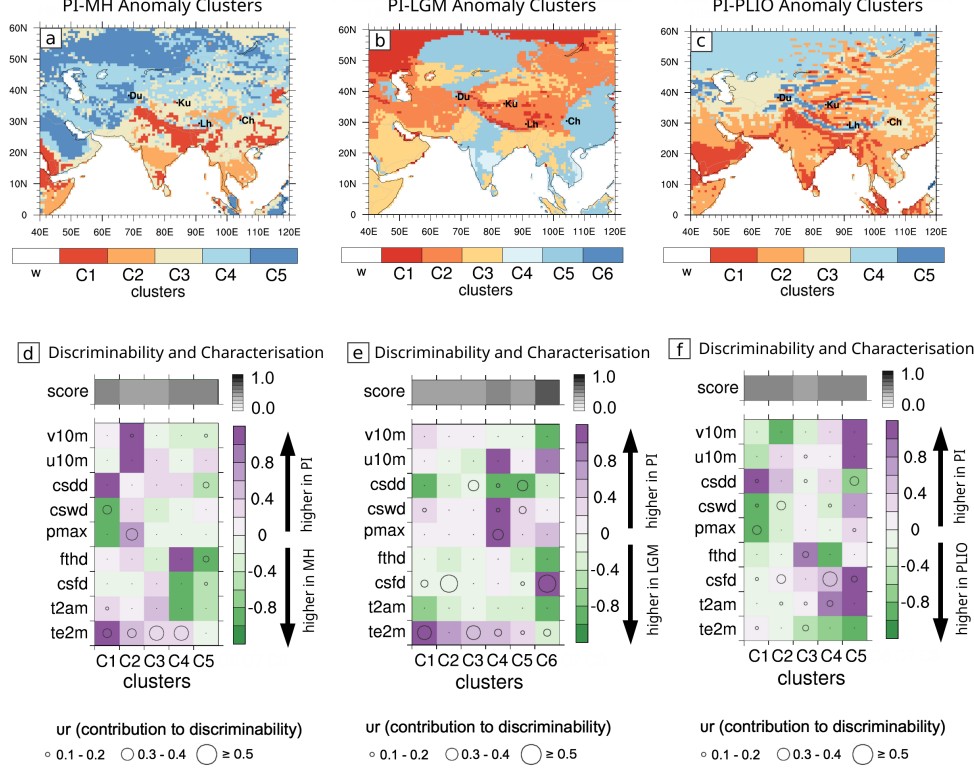

Fig. 5 The multivariate anomaly maps for time slice comparisons PI-MH(a), PI-LGM(b) and PI-PLIO(c) show the geographical coverage of clusters $C_1$-$C_i$ in Himalaya-Tibet, which describe the spatial extend of regions characterised by similar mod of change. The corresponding modes of change (d,e and f) for each cluster are expressed as relative changes in each of the 9 investigated variables (table 1): 2m air temperature (te2m), 2m air temperature amplitude (t2am), consecutive freezing days (csfd), freeze-thaw days (fthd), maximum precipitation (pmax), consecutive wet days (cswd), consecutive dry days (csdd), zonal near surface wind speeds (u10) and meridional near surface wind speeds (v10). The score (d,e and f) expresses the goodness of discriminability between the palaeoclimate pairs PI-MH(d), PI-LGM(e) and PI-PLIO(f) in each of the anomaly clusters. The size of the circles corresponds to the relative contribution of each of the 9 climatic attribute variables to the measured discriminability in each anomaly cluster for all three time slice comparisons.








| Attribute Variables | | | |
|---|---|---|---|
| Code | Units | Explanation | Geomorphic relevance |
| te2m | °C | Mean annual air temperature at 2m height | (peri-)glacial processes, vegetation |
| t2am | °C | Maximum intra-monthly variation of 2m air temperature | (peri-)glacial processes, vegetation, frost-cracking |
| csfd | days | Number of consecutive days with surface temperature conditions below 0°C | (peri-)glacial processes, frost cracking |
| fthd | days | Number of times the 0°C threshold is crossed from day to day | (peri-)glacial processes, frost cracking |
| pmax | mm/d | Maximum daily precipitation value in a month | landslides, runoff, river incision, vegetation conditions |
| cswd | days | Number of consecutive days experiencing precipitation exceeding 1 mm/d | landslides, runoff, river incision, vegetation conditions |
| csdd | days | Number of consecutive days experiencing precipitation below 1 mm/d | landslides, runoff, river incision, vegetation, aeolian erosion |
| u10 | m/s | Zonal (along-latitude) wind speeds at 10m height | aeolian erosion, transport, deposition, raindrop trajectories |
| v10 | m/s | Zonal (along-longitude) wind speeds at 10m height | aeolian erosion, transport, deposition, raindrop trajectories |

Table 1: Code, units, explanation and geomorphic relevance of each of the climate attribute variables used in the
cluster- and discriminant analysis.