# Peer review of "Detection and Explanation of Spatiotemporal Patterns in Late Cenozoic Palaeoclimate Change Relevant to Earth Surface Processes"

_Earth Surface Dynamics, 2019_

## Referee Comment (RC1) · Anonymous Referee #1 · 7 Apr 2019

Review of the manuscript entitled "Detection and Explanation of Spatiotemporal Patterns in Late Cenozoic Palaeoclimate Change Relevant to Earth Surface Processes" submitted to Earth System Dynamics Discussions by Sebastian G. Mutz and Todd. A. Ehlers.

Overall rating: to be accepted with minor revisions General Comments: the manuscript deals in a very concise way with the description and application of a methodology that combines geographical clustering with discrimination analysis, to detect and explain differences in different past climates. The study is in general thorough and very well

structured.

Scientific significance: the study shows a methodology applied to the issue of climate change induced changes in Earth surface processes. It does not get entirely clear which part of the methodology can be considered as original to this manuscript. It would be good to clarify that. But generally speaking this methodology and the way how the results are interpreted and illustrated is a substantial contribution for the scientific community and fits to the topics of ESD.

Scientific Quality: the methodology is valid and robust and the results are well discussed. The results are discussed against the broader picture of state of the art knowledge of climatic differences between the respective time slices. References are appropriate. Some additional plots giving valuable information for the evaluation of the results are found in the supplementary material.

Presentation Quality: the text is very well structured and the methodology and the results are very clearly explained in detail. The figures are clear and very well readable, very well structured and the figure captions do extensively explain details. The amount of figures is sufficient. I'm not a native speaker, but according to my knowledge the use of the English language is appropriate.

Minor comments:

- Abstract: in the abstract the authors mention tipping points in what I consider the motivation of the research. This, in my opinion is misleading (even though the method might be used for that purpose) since the rest of the manuscript is not about tipping points. Better mention the effects on land surface processes like erosion already as motivation in the abstract (like it is done nicely in the introduction).

- For the motivation, as well as in the discussion section it could be mentioned that this methodology might be pretty useful regarding the effects of future climate changes and the impacts on Earth surface processes.

- Introduction line 70: Please do not call modern pre-industrial, since there is distinct differences and this might lead to confusions

- Introduction lines 79/80: it is not entirely clear to me why you mention uplift histories here. Could you maybe clarify that? Could that methodology help to re-evaluate uplift histories?

- Paragraph 2.3./line 202: add "climate" to "Each variable" => "Each climate variable"

- Line 230: given by the models

- Line 461: impacts on physical weathering

- Discussion: please mention more often in the discussion where the different named regions are situated (e.g. mountains, lowlands etc).

- Conclusion: line 543: please do not only mention k, but also what it means (climate anomaly clusters?)  here, since there is people around reading only the conclusions and the abstract. . ..

- Line556: . . .coincide with decreases in . . ...

- Conclusion: It would be good to mention future applicability of the method like using it for the investigation of other physical processes or other time slices (give examples!).

---

## Referee Comment (RC2) · Andrew Wickert (Referee) · 17 Apr 2019

Mutz and Ehlers here present a description of their recent efforts to link GCM outputs with surface processes and landscape evolution. They analyze regions, or "clusters", across which they observe characteristic sets of changes between pre-industrial time and different times in the past. They have also published their model outputs, ensuring that the present analysis is not the only one that may be performed. Aside from a few points where I was not able to follow the argument, I found the paper to be well-organized and appropriate for publication. I feel that this paper should be accepted

with minor revisions, according to the line-by-line comments below.

Line-by-line:

16, 21, etc. I think that there should be emdashes instead of hyphens between PI and MH, LGM, and PLIO

24. explained –> described.

48–49. Modern climate is not always assumed, though. As an example with which I am quite familiar, we (Wickert et al., 2019) actually used plausible river discharges based on glacial-stage conditions including ice-sheet melt and precipitation minus evapotranspiration (Wickert, 2016) to understand incision of the Mississippi River. I strongly agree that using the appropriate past climate is important! But I am wondering if "often have to be interpreted" is a bit too strong in general – from paleo-proxy data, we generally do know something about how climate varied. But perhaps what you mean to say is that there is not a globally-spanning resource to do this and ingest it into models, which could in fact be closer to your aim (?) and a stronger way to state this.

p.2 and 3 overall: Consider ways to split this 1.5-page single paragraph – perhaps between background and what you do. (Even if it is supposed to be split at the list and I just don't see it, the p. 2 single paragraph definitely could benefit from being broken up into its sub-topics.

79. "lies on" seems awkward to me. Why not, "We focus on 4 regions..."

93, 95. Earth-surface here is a compound noun, and so should be hyphenated. It also seems that 93-95 is an accidental self-standing one-sentence paragraph

99. surface-process-relevant (compound noun)

112. Consider updating the PMIP reference, as new PMIP projects have run since 2012 – we are now on PMIP4! The following reference is an experiment overview. But in addition to just citing, I think that their science and model outputs could be quite

useful to you and your future work.

Kageyama, M., Braconnot, P., Harrison, S. P., Haywood, A. M., Jungclaus, J. H., Otto-Bliesner, B. L., ... Zhou, T. (2018). The PMIP4 contribution to CMIP6 – Part 1: Overview and over-arching analysis plan. Geoscientific Model Development, 11(3), 1033–1057. https://doi.org/10.5194/gmd-11-1033-2018

119. Consider starting a new paragraph at "Mutz"

142. "of interest to the Earth surface": rephrase. I know what you mean, but Earth's surface doesn't have the mental capacity for interests.

143. communities (plural)

144-145. hyphens to emdashes (indicating range)

163-170. In your place, I would have added evapotranspiration (or P-ET) to this list, as the sum of latter across watershed is what should control river discharge. Could you explain your decision not to include this?

175. New paragraph at "clustering".

178-180. I think that the writing would be clearer if you focus on this criterion before the preceding sentence, in which you give the appropriate values. So, methods → results.

203. I'm a bit confused here by the mention of the number of cluster members. I had thought that the clusters themselves were not yet defined. Or by this, do you mean "grid boxes"? Or "groups selected and then tested"?

Equation 4 seems unnecessary – perhaps you can just write that you maximized $\Gamma$.

Equation 6 seems strange for multiple reasons. But now I realize that what looks like strange matrix multiplication is actually really several separate matrices. Please also ensure that you have reasonable formats, including consistent positioning and orientation of ellipses, brackets that look like brackets long brackets (this is more for

typesetting, but these look like you just stretched short brackets, thus ending up with long knobs on the ends), etc. The latter comments are for style, but the former are for content.

224. Could you describe your solution technique in more detail? I do not understand why you needed to partially differentiate the equation (or with respect to what) in order to solve it.

224. Could you explain mathematically what you mean by "standardized"? This is the point at which I no longer have enough information to understand or reproduce your approach. In addition, I quite dislike two-letter variables: it is then quite difficult to know if they represent two variables that should be multiplied together, or a single variable.

253. Please read this paragraph and split it at the point at which "Erosion rates..." is no longer a good topic sentence. I suggest at, "The anomaly cluster".

262-263. The "Late Pliocene" sentence seems a bit unnecessary – if you want to add an example, consider using something with a reference to demonstrate that this would be the case.

316. Argentina, AND THE central and...

364. endashes

364. "decreases in 2m air temperature AND freeze-thaw days, and": separate decreases and increases. In general, I suggest that you use the Oxford comma; it can disambiguate your complex lists.

366. These cells are covered by more than just the Scandinavian ice sheet.

399. PLIO→MH?

423. incl. The→including the

424. temperature, and WITH (required to split lists)

458-459. large-scale

462. and→as well as to (required to split lists)

490-494. Consider noting something about changes in evapotranspiration vs. changes in erosion rate due to the physical presence of vegetation (e.g., roots)

525. erosion-relevant

525. By "high" do you mean small spatial, short temporal, both, neither?

Data availability: I suggest that you obtain a DOI for your model outputs.

---

## Author Comment (AC1) · 22 May 2019

**Response to RC1 (Anonymous Referee 1)**

We thank the anonymous referee for the time and effort they spent on reading, reviewing and commenting our manuscript. We were able to improve the manuscript by following their suggestions and thinking and acting on their comments. In this communication, we highlight our responses in blue.

It does not get entirely clear which part of the methodology can be considered as original to this manuscript. It would be good to clarify that.

The individual components of the whole procedure are established methods, but the combination of methods was tailored specifically to the scientific problem at hand. We made sure to clarify this in the methods section (L136-138), and mention examples of where else this chain of processing may be used in the discussion (L677-687) .

Minor comments:
- Abstract: in the abstract the authors mention tipping points in what I consider the motivation of the research. This, in my opinion is misleading (even though the method might be used for that purpose) since the rest of the manuscript is not about tipping points. Better mention the effects on land surface processes like erosion already as motivation in the abstract (like it is done nicely in the introduction).

We thank the reviewer for noticing and highlighting this inconsistency and agree. We now omit the mention of tipping points and focus the reader on surface processes/erosion from the beginning.

- For the motivation, as well as in the discussion section it could be mentioned that this methodology might be pretty useful regarding the effects of future climate changes and the impacts on Earth surface processes.

This is an excellent point and may remove barriers for climate impact researchers to apply a similar method for future climate change. We make a note of that in our manuscript now (L125-127 and L677-678).

- Introduction line 70: Please do not call modern pre-industrial, since there is distinct differences and this might lead to confusions

We thank the reviewer for raising this very valid point and made sure we do not mix those two up throughout the entire manuscript.

- Introduction lines 79/80: it is not entirely clear to me why you mention uplift histories here. Could you maybe clarify that? Could that methodology help to re-evaluate uplift histories?

We thank the reviewer for pointing this out. We originally mentioned uplift histories, because reconstructions and interpretations of erosion histories impact the refinement of uplift histories. We realise now that the mention of uplift histories is confusing at this point in the manuscript and omitted it.

- Paragraph 2.3./line 202: add "climate" to "Each variable" => "Each climate variable"
This has been corrected.

- Line 230: given by the models
This has been corrected.

- Line 461: impacts on physical weathering
This has been corrected.

\- Discussion: please mention more often in the discussion where the different named regions are situated (e.g. mountains, lowlands etc).
We added more such descriptive terms to the discussion when we mention specific regions/clusters (e.g. L554, 576, 590).

\- Conclusion: line 543: please do not only mention k, but also what it means (climate anomaly lusters?) here, since there is people around reading only the conclusions and the abstract:
This is a very good point. We instead write "assigned number of anomaly clusters" (L659), but still mention k in brackets to help those who have read the entire manuscript make that connection more easily.

\- Line556: coincide with decreases in
This has been corrected.

\- Conclusion: It would be good to mention future applicability of the method like using it for the investigation of other physical processes or other time slices (give examples!).
We thank the reviewer for the suggestion. We mention at the end of the discussion that the presented methods may be applied to future climates. Additionally, we give a possible example of method transfer to a different research field and highlight the important conditions of merited use of the method. (L677-687)

---

## Author Comment (AC2) · 22 May 2019

**Response to RC2 (Andrew Wickert)**

We thank Andrew Wickert for his time and effort as a referee. We appreciate his thoroughness, comments and suggestions and think we were able to improve our manuscript by acting on those. In this communication, we highlight our responses in blue.

16, 21, etc. I think that there should be emdashes instead of hyphens between PI and MH, LGM, and PLIO
This has been corrected throughout the entire manuscript.

24. explained –> described.
We use the term "explain" in a statistical sense, as is consistent with the language used to describe the results of the discriminant analysis. The LMD is an explanatory tool. However, we acknowledge that this may create some confusion in the abstract. We therefore now explicitly state in the abstract that this explanation is a statistical one.

48–49. Modern climate is not always assumed, though. As an example with which I am quite familiar, we (Wickert et al., 2019) actually used plausible river discharges based on glacial-stage conditions including ice-sheet melt and precipitation minus evapotranspiration (Wickert, 2016) to understand incision of the Mississippi River. I strongly agree that using the appropriate past climate is important! But I am wondering if "often have to be interpreted" is a bit too strong in general – from paleo-proxy data, we generally do know something about how climate varied. But perhaps what you mean to say is that there is not a globally-spanning resource to do this and ingest it into models, which could in fact be closer to your aim (?) and a stronger way to state this. p.2 and 3 overall: Consider ways to split this 1.5-page single paragraph – perhaps between background and what you do. (Even if it is supposed to be split at the list and I just don't see it, the p. 2 single paragraph definitely could benefit from being broken up into its sub-topics.
We thank Andrew Wickert for this comment. Indeed, we could have been clearer in this regard. The advantages of GCM's are (1) having a globally-spanning resource, and (2) a way to relate observations of palaeoclimate back to processes and drivers, since GCM's are based on our physical understanding of the climate system, (3) they are a tool suitable for sensitivity experiments, (4) they complement proxy-based reconstructions with regional averages and additional climate variables to offer a more complete picture of climate. We agree that our original phrasing is a little strong. We adjusted this and offer a clearer motivation for using GCM's for these purposes. Additionally, we split the paragraph intro three parts: (1) General motivation for the work and using GCM output, (2) previous palaeoclimate modelling work, and (3) what we do here. (L54-64)

79. "lies on" seems awkward to me. Why not, "We focus on 4 regions..."
Many thanks for the suggestion. We changed this in our text.

93, 95. Earth-surface here is a compound noun, and so should be hyphenated.
This has been corrected.

It also seems that 93-95 is an accidental self-standing one-sentence paragraph
This has been corrected.

99. surface-process-relevant (compound noun)
This has been corrected.

112. Consider updating the PMIP reference, as new PMIP projects have run since 2012 – we are now on PMIP4! The following reference is an experiment overview. But in addition to just citing, I think that their science and model outputs could be quite useful to you and your future work.
Kageyama, M., Braconnot, P., Harrison, S. P., Haywood, A. M., Jungclaus, J. H., Otto-Bliesner, B. L., ... Zhou, T. (2018). The PMIP4 contribution to CMIP6 – Part 1: Overview

and over-arching analysis plan. Geoscientific Model Development, 11(3), 1033–1057.
https://doi.org/10.5194/gmd-11-1033-2018

We updated the reference in the text, but still included the Bracannot reference since our simulations were based on simulations and protocols from the third phase. We agree that PMIP4 will be very relevant to us and certainly plan to take advantage of PMIP4 in future work.

119. Consider starting a new paragraph at "Mutz"
We agree that this gives the text more structure and followed the suggestion.

142. "of interest to the Earth surface": rephrase. I know what you mean, but Earth's surface doesn't have the mental capacity for interests.
We added a hyphen to clarify that we speak of the Earth-surface communities (and palaeoaltimetry communities).

143. communities (plural)
This has been corrected.

144-145. hyphens to emdashes (indicating range)
This has been corrected.

163-170. In your place, I would have added evapotranspiration (or P-ET) to this list, as the sum of latter across watershed is what should control river discharge. Could you explain your decision not to include this?
We refrained from using several potentially relevant variables due to their poor representation in GCM's on the scale we are working on. GCM's don't represent the regional hydrology well due to the parameterisations in their land surface schemes. Routing GCM's through more sophisticated hydrological models or regional climate models would allow the inclusion of such variables. We changed the text to be clearer about our choice of variables (in "Data and Methods"), and added those limitations (and suggestions) to the "Comments on methodical implications" section.

175. New paragraph at "clustering".
This has been corrected.

178-180. I think that the writing would be clearer if you focus on this criterion before the preceding sentence, in which you give the appropriate values. So, methods ! results.
We re-arranged these sentences, so that the order of information makes more sense.

203. I'm a bit confused here by the mention of the number of cluster members. I had thought that the clusters themselves were not yet defined. Or by this, do you mean "grid boxes"? Or "groups selected and then tested"?
We thank Andrew Wickert for pointing out the confusing nature of this sentence. As summed up in Fig. 1, we carry out the clustering (and find the optimal cluster number k) before we conduct the LMD. However, we refer here to the number of cluster members, i.e. the number of elements or grid boxes in each cluster. We realise now that we have not used the term "cluster members" before and replaced it by "elements in each cluster" to avoid any such confusion.

Equation 4 seems unnecessary – perhaps you can just write that you maximized.
We acknowledge the redundancy and overlap between equation 3 and 4. However, we think equation 4 nicely summarises the whole problem in a way that you also may find it summarised in text books, thus making it easier to recognise the same thing again when reading up on LMD. We therefore decided to leave out equation 3 and keep equation 4 (now 3).

Equation 6 seems strange for multiple reasons. But now I realize that what looks like strange matrix multiplication is actually really several separate matrices. Please also ensure that you have reasonable formats, including consistent positioning and orientation of ellipses, brackets that look like brackets

long brackets (this is more for typesetting, but these look like you just stretched short brackets, thus ending up with long knobs on the ends), etc. The latter comments are for style, but the former are for content.

6 (now 5 in the revised manuscript) merely expands and explains the terms of the previous equation, which is a common notation to shorten the notation for multiple regressions (cf. next comment response). It may not be necessary to include this to expand on what is already stated before, but we hope that this will look more familiar to some of the readers. We reworked the formatting.

224. Could you describe your solution technique in more detail? I do not understand why you needed to partially differentiate the equation (or with respect to what) in order to solve it.
It would involve partial differentiation with respect to discriminant coefficients, which can be seen as regression coefficients if we think of it as a multiple regression for the elements on the discrimination axis. We clarified this and inserted a reference for more details, since this is a well-established method.

224. Could you explain mathematically what you mean by "standardized"? This is the point at which I no longer have enough information to understand or reproduce your approach. In addition, I quite dislike two-letter variables: it is then quite difficult to know if they represent two variables that should be multiplied together, or a single variable.
We thank Andrew Wickert for these comments. We use the term "standardised" in a statistical sense, i.e. we put the value in relation to the standard deviation of the respective variable. We clarified this. We agree that using two-letter variables can lead to confusion. We changed the variable names to one-letter variables and updated them throughout the manuscript and on our figures.

253. Please read this paragraph and split it at the point at which "Erosion rates..." is no longer a good topic sentence. I suggest at, "The anomaly cluster".
We agree that this gives the text more structure and followed the suggestion.

262-263. The "Late Pliocene" sentence seems a bit unnecessary – if you want to add an example, consider using something with a reference to demonstrate that this would be the case.
We decided to remove this sentence as it is somewhat distracting and unnecessary.

316. Argentina, AND THE central and...
This has been corrected.

364. endashes
This has been corrected.

364. "decreases in 2m air temperature AND freeze-thaw days, and": separate decreases and increases. In general, I suggest that you use the Oxford comma; it can disambiguate your complex lists.
We appreciate the suggestion and made changes accordingly.

366. These cells are covered by more than just the Scandinavian ice sheet.
We updated the text to make sure we include the British Ice Sheet.

399. PLIO!MH?
Yes, this should read "MH". We corrected it.

423. incl. The!including the
This has been corrected.

424. temperature, and WITH (required to split lists)
This has been corrected.

458-459. large-scale
This has been corrected.

462. and→as well as to (required to split lists)
This has been corrected.

490-494. Consider noting something about changes in evapotranspiration vs. changes in erosion rate due to the physical presence of vegetation (e.g., roots)
We have included a note on vegetation's modification of the hydrological cycle, and modifying hillslope stability by changing root characteristics.

525. erosion-relevant
This has been corrected.

525. By "high" do you mean small spatial, short temporal, both, neither?
We mean both and clarify this now in the text.

Data availability: I suggest that you obtain a DOI for your model outputs.
As we understand, it is not possible to obtain a doi for self-hosted data, and the size of the original model output is currently still too large for hosting services.

---

## Author Response (AR2)

**Response to Reviewer Comments**

**Response to RC1 (Anonymous Referee 1)**

We thank the anonymous referee for the time and effort they spent on reading, reviewing and commenting our manuscript. We were able to improve the manuscript by following their suggestions and thinking and acting on their comments. In this communication, we highlight our responses in blue.

It does not get entirely clear which part of the methodology can be considered as original to this manuscript. It would be good to clarify that.

The individual components of the whole procedure are established methods, but the combination of methods was tailored specifically to the scientific problem at hand. We made sure to clarify this in the methods section (L136-138), and mention examples of where else this chain of processing may be used in the discussion (L677-687) .

Minor comments:
- Abstract: in the abstract the authors mention tipping points in what I consider the motivation of the research. This, in my opinion is misleading (even though the method might be used for that purpose) since the rest of the manuscript is not about tipping points. Better mention the effects on land surface processes like erosion already as motivation in the abstract (like it is done nicely in the introduction).

We thank the reviewer for noticing and highlighting this inconsistency and agree. We now omit the mention of tipping points and focus the reader on surface processes/erosion from the beginning.

- For the motivation, as well as in the discussion section it could be mentioned that this methodology might be pretty useful regarding the effects of future climate changes and the impacts on Earth surface processes.

This is an excellent point and may remove barriers for climate impact researchers to apply a similar method for future climate change. We make a note of that in our manuscript now (L125-127 and L677-678).

- Introduction line 70: Please do not call modern pre-industrial, since there is distinct differences and this might lead to confusions

We thank the reviewer for raising this very valid point and made sure we do not mix those two up throughout the entire manuscript.

- Introduction lines 79/80: it is not entirely clear to me why you mention uplift histories here. Could you maybe clarify that? Could that methodology help to re-evaluate uplift histories?

We thank the reviewer for pointing this out. We originally mentioned uplift histories, because reconstructions and interpretations of erosion histories impact the refinement of uplift histories. We realise now that the mention of uplift histories is confusing at this point in the manuscript and omitted it.

- Paragraph 2.3./line 202: add "climate" to "Each variable" => "Each climate variable"
This has been corrected.

- Line 230: given by the models

This has been corrected.

- Line 461: impacts on physical weathering
This has been corrected.

- Discussion: please mention more often in the discussion where the different named regions are situated (e.g. mountains, lowlands etc).
We added more such descriptive terms to the discussion when we mention specific regions/clusters (e.g. L554, 576, 590).

- Conclusion: line 543: please do not only mention k, but also what it means (climate anomaly lusters?) here, since there is people around reading only the conclusions and the abstract:
This is a very good point. We instead write "assigned number of anomaly clusters" (L659), but still mention k in brackets to help those who have read the entire manuscript make that connection more easily.

- Line556: coincide with decreases in
This has been corrected.

- Conclusion: It would be good to mention future applicability of the method like using it for the investigation of other physical processes or other time slices (give examples!).
We thank the reviewer for the suggestion. We mention at the end of the discussion that the presented methods may be applied to future climates. Additionally, we give a possible example of method transfer to a different research field and highlight the important conditions of merited use of the method. (L677-687)

**Response to RC2 (Andrew Wickert)**

We thank Andrew Wickert for his time and effort as a referee. We appreciate his thoroughness, comments and suggestions and think we were able to improve our manuscript by acting on those. In this communication, we highlight our responses in blue.

16, 21, etc. I think that there should be emdashes instead of hyphens between PI and MH, LGM, and PLIO
This has been corrected throughout the entire manuscript.

24. explained –> described.
We use the term "explain" in a statistical sense, as is consistent with the language used to describe the results of the discriminant analysis. The LMD is an explanatory tool. However, we acknowledge that this may create some confusion in the abstract. We therefore now explicitly state in the abstract that this explanation is a statistical one.

48–49. Modern climate is not always assumed, though. As an example with which I am quite familiar, we (Wickert et al., 2019) actually used plausible river discharges based on glacial-stage conditions including ice-sheet melt and precipitation minus evapotranspiration (Wickert, 2016) to understand incision of the Mississippi River. I strongly agree that using the appropriate past climate is important! But I am wondering if "often have to be interpreted" is a bit too strong in general – from paleo-proxy data, we generally do know something about how climate varied. But perhaps what you mean to say is that there is not a globally-spanning resource to do this and ingest it into models, which could in fact be closer to your aim (?) and a stronger way to state this. p.2 and 3 overall: Consider ways to split this 1.5-page single paragraph – perhaps between background and what you do. (Even if it is supposed to be split at the list and I just don't see it, the p. 2 single paragraph definitely could benefit from being broken up into its sub-topics.)
We thank Andrew Wickert for this comment. Indeed, we could have been clearer in this regard. The advantages of GCMs are (1) having a globally-spanning resource, and (2) a way to relate observations of palaeoclimate back to processes and drivers, since GCMs are based on our physical  understanding of the climate system, (3) they are a tool suitable for sensitivity experiments, (4) they complement proxy-based reconstructions with regional averages and additional climate variables to offer a more complete picture of climate. We agree that our original phrasing is a little strong. We adjusted this and offer a clearer motivation for using GCMs for these purposes. Additionally, we split the paragraph intro three parts: (1) General motivation for the work and using GCM output, (2) previous palaeoclimate modelling work, and (3) what we do here. (L54-64)

79. "lies on" seems awkward to me. Why not, "We focus on 4 regions..."
Many thanks for the suggestion. We changed this in our text.

93, 95. Earth-surface here is a compound noun, and so should be hyphenated.
This has been corrected.

It also seems that 93-95 is an accidental self-standing one-sentence paragraph
This has been corrected.

99. surface-process-relevant (compound noun)
This has been corrected.

112. Consider updating the PMIP reference, as new PMIP projects have run since 2012 – we are now on PMIP4! The following reference is an experiment overview. But in addition to just citing, I think that their science and model outputs could be quite useful to you and your future work.
Kageyama, M., Braconnot, P., Harrison, S. P., Haywood, A. M., Jungclaus, J. H., Otto-

| | |
|---|---|
| Deleted: ' | |
| Deleted: ' | |
| Deleted: ' | |

Bliesner, B. L., ... Zhou, T. (2018). The PMIP4 contribution to CMIP6 – Part 1: Overview and over-arching analysis plan. Geoscientific Model Development, 11(3), 1033–1057. https://doi.org/10.5194/gmd-11-1033-2018

We updated the reference in the text, but still included the Bracannot reference since our simulations were based on simulations and protocols from the third phase. We agree that PMIP4 will be very relevant to us and certainly plan to take advantage of PMIP4 in future work.

119. Consider starting a new paragraph at "Mutz"
We agree that this gives the text more structure and followed the suggestion.

142. "of interest to the Earth surface": rephrase. I know what you mean, but Earth's surface doesn't have the mental capacity for interests.
We added a hyphen to clarify that we speak of the Earth-surface communities (and palaeoaltimetry communities).

143. communities (plural)
This has been corrected.

144-145. hyphens to emdashes (indicating range)
This has been corrected.

163-170. In your place, I would have added evapotranspiration (or P-ET) to this list, as the sum of latter across watershed is what should control river discharge. Could you explain your decision not to include this?
We refrained from using several potentially relevant variables due to their poor representation in GCMs on the scale we are working on. GCMs don't represent the regional hydrology well due to the parameterisations in their land surface schemes. Routing GCMs through more sophisticated hydrological models or regional climate models would allow the inclusion of such variables. We changed the text to be clearer about our choice of variables (in "Data and Methods"), and added those limitations (and suggestions) to the "Comments on methodical implications" section.

175. New paragraph at "clustering".
This has been corrected.

178-180. I think that the writing would be clearer if you focus on this criterion before the preceding sentence, in which you give the appropriate values. So, methods ! results.
We re-arranged these sentences, so that the order of information makes more sense.

203. I'm a bit confused here by the mention of the number of cluster members. I had thought that the clusters themselves were not yet defined. Or by this, do you mean "grid boxes"? Or "groups selected and then tested"?
We thank Andrew Wickert for pointing out the confusing nature of this sentence. As summed up in Fig. 1, we carry out the clustering (and find the optimal cluster number k) before we conduct the LMD. However, we refer here to the number of cluster members, i.e. the number of elements or grid boxes in each cluster. We realise now that we have not used the term "cluster members" before and replaced it by "elements in each cluster" to avoid any such confusion.

Equation 4 seems unnecessary – perhaps you can just write that you maximized.
We acknowledge the redundancy and overlap between equation 3 and 4. However, we think equation 4 nicely summarises the whole problem in a way that you also may find it summarised in text books, thus making it easier to recognise the same thing again when reading up on LMD. We therefore decided to leave out equation 3 and keep equation 4 (now 3).

Equation 6 seems strange for multiple reasons. But now I realize that what looks like strange matrix multiplication is actually really several separate matrices. Please also ensure that you have reasonable

210 formats, including consistent positioning and orientation of ellipses, brackets that look like brackets long brackets (this is more for typesetting, but these look like you just stretched short brackets, thus ending up with long knobs on the ends), etc. The latter comments are for style, but the former are for content.
6 (now 5 in the revised manuscript) merely expands and explains the terms of the previous equation, which is a common notation to shorten the notation for multiple regressions (cf. next comment
215 response). It may not be necessary to include this to expand on what is already stated before, but we hope that this will look more familiar to some of the readers. We reworked the formatting.

224. Could you describe your solution technique in more detail? I do not understand why you needed to partially differentiate the equation (or with respect to what) in order to solve it.
220 It would involve partial differentiation with respect to discriminant coefficients, which can be seen as regression coefficients if we think of it as a multiple regression for the elements on the discrimination axis. We clarified this and inserted a reference for more details, since this is a well-established method.

225 224. Could you explain mathematically what you mean by "standardized"? This is the point at which I no longer have enough information to understand or reproduce your approach. In addition, I quite dislike two-letter variables: it is then quite difficult to know if they represent two variables that should be multiplied together, or a single variable.
We thank Andrew Wickert for these comments. We use the term "standardised" in a statistical sense,
230 i.e. we put the value in relation to the standard deviation of the respective variable. We clarified this. We agree that using two-letter variables can lead to confusion. We changed the variable names to one-letter variables and updated them throughout the manuscript and on our figures.

253. Please read this paragraph and split it at the point at which "Erosion rates..." is no
235 longer a good topic sentence. I suggest at, "The anomaly cluster".
We agree that this gives the text more structure and followed the suggestion.

262-263. The "Late Pliocene" sentence seems a bit unnecessary – if you want to add an example, consider using something with a reference to demonstrate that this would be the case.
240 We decided to remove this sentence as it is somewhat distracting and unnecessary.

316. Argentina, AND THE central and...
This has been corrected.

245 364. endashes
This has been corrected.

364. "decreases in 2m air temperature AND freeze-thaw days, and": separate decreases and increases. In general, I suggest that you use the Oxford comma; it can
250 disambiguate your complex lists.
We appreciate the suggestion and made changes accordingly.

366. These cells are covered by more than just the Scandinavian ice sheet.
We updated the text to make sure we include the British Ice Sheet.
255
399. PLIO!MH?
Yes, this should read "MH". We corrected it.

423. incl. The!including the
260 This has been corrected.

424. temperature, and WITH (required to split lists)
This has been corrected.

458-459. large-scale
This has been corrected.

462. and→as well as to (required to split lists)
This has been corrected.

490-494. Consider noting something about changes in evapotranspiration vs. changes in erosion rate
due to the physical presence of vegetation (e.g., roots)
We have included a note on vegetation's modification of the hydrological cycle, and modifying
hillslope stability by changing root characteristics.

525. erosion-relevant
This has been corrected.

525. By "high" do you mean small spatial, short temporal, both, neither?
We mean both and clarify this now in the text.

Data availability: I suggest that you obtain a DOI for your model outputs.
As we understand, it is not possible to obtain a doi for self-hosted data, and the size of the original
model output is currently still too large for hosting services.

**List of Relevant Changes**

305

Following the suggestions of the reviewers, we clarified many details regarding our method, emphasised the motivation of a GCM (General Circulation Model) based study, gave the manuscript more structure and clarity overall, listed further limitation and suggestions, and highlighted possibilities of method transfer. Furthermore, we updated our figures to accommodate a suggestion for change in notation. Finally, we followed the technical

310    suggestions by the editor.

315

320

325

330

**Detection and Explanation of Spatiotemporal Patterns in Late Cenozoic Palaeoclimate Change Relevant to Earth Surface Processes**

Sebastian G. Mutz[1], Todd A. Ehlers[1]

[1] Department of Geosciences, University Tübingen, D-72074 Tübingen, Germany

*Correspondence to*: Sebastian G. Mutz (sebastian.mutz@uni-tuebingen.de)

**Abstract**

Detecting and explaining differences between palaeoclimates can provide valuable insights for Earth scientists investigating processes that are affected by climate change over geologic time. In this study, we describe and explain spatiotemporal patterns in palaeoclimate change that are relevant to Earth surface scientists. We apply a combination of multivariate cluster and discriminant analysis techniques to a set of high-resolution palaeoclimate simulations. The simulations were conducted with the ECHAM5 climate model and consistent set-up. A pre-industrial (PI) climate simulation serves as the control experiment, which is compared to a suite of simulations of Late Cenozoic climates, namely a Mid-Holocene (MH, ca. 6.5 ka), Last Glacial Maximum (LGM, ca. 21 ka) and Pliocene (PLIO, ca. 3 Ma) climate. For each of the study regions (Western South America, Europe, Himalaya-Tibet and South Alaska), differences in climate are subjected to geographical clustering to identify dominant modes of climate change and their spatial extent for each time slice comparison (PI–MH, PI–LGM and PI–PLIO). The selection of climate variables for the cluster analysis is made on the basis of their relevance to Earth surface processes and includes 2m air temperature, 2m air temperature amplitude, consecutive freezing days, freeze-thaw days, maximum precipitation, consecutive wet days, consecutive dry days, zonal wind speed and meridional wind speed. We then apply a two-class multivariate discriminant analysis to simulation pairs PI–MH, PI–LGM and PI–PLIO to evaluate and explain the discriminability between climates within each of the anomaly clusters. Changes in ice cover create the most distinct and stable patterns of climate change, and create the best discriminability between climates in western Patagonia. The distinct nature of European palaeoclimates is statistically explained mostly by changes in 2m air temperature (MH, LGM, PLIO), consecutive freezing days (LGM) and consecutive wet days (PLIO). These factors typically contribute 30%-50%, 10%-40% and 10%-30% respectively to climate discriminability. Finally, our results identify regions particularly prone to changes in precipitation-induced erosion and temperature-dependent physical weathering.

**Keywords:** Cenozoic climate, climate change, ECHAM5, Last Glacial Maximum, Mid-Holocene, Pliocene, discriminant analysis, discriminability, Earth surface processes, erosion

**1. Introduction**

In the study of Earth surface processes, gaining new quantitative understanding of the atmosphere's interaction with the Earth's surface through erosional processes is limited by the difficulty of establishing reliable palaeoclimatic context for erosion rate histories. Such context is particularly useful when erosion rates are calculated using techniques such as cosmogenic radionuclides and low-temperature thermochronology [e.g. Schaller et al., 2002; Bookhagen et al., 2005; Moon et al., 2011; Insel et al., 2010; Stock et al., 2009], which

**Commented [SM1]:** Since we work with simulations we previously conducted, we believe it would be misleading to follow the suggestion exactly. Instead, we tried to avoid long strings of adjectives by rephrasing it as such.

[revised manuscript text omitted]